# A Photosensitivity-Enhanced Plant Growth Algorithm for UAV Path Planning

**DOI:** 10.3390/biomimetics9040212

**Published:** 2024-03-31

**Authors:** Renjie Yang, Pan Huang, Hui Gao, Qingyang Qin, Tao Guo, Yongchao Wang, Yaoming Zhou

**Affiliations:** 1School of Aeronautic Science and Engineering, Beihang University, Beijing 100191, China; yangrenjie@buaa.edu.cn (R.Y.); gaohui@buaa.edu.cn (H.G.); qinqingyang@buaa.edu.cn (Q.Q.); guotao0320@buaa.edu.cn (T.G.); 2School of Astronautics, Beihang University, Beijing 100083, China; by2315144@buaa.edu.cn; 3Research Institute of Intelligent Decision Engineering, CASIC, Wuhan 430040, China; 4College of Control Science and Engineering, Zhejiang University, Hangzhou 310023, China; 5Tianmushan Laboratory, Hangzhou 310023, China

**Keywords:** unmanned aerial vehicle, path planning, plant growth, heuristic information, electronic chart

## Abstract

With the rise and development of autonomy and intelligence technologies, UAVs will have increasingly significant applications in the future. It is very important to solve the problem of low-altitude penetration of UAVs to protect national territorial security. Based on an S-57 electronic chart file, the land, island, and threat information for an actual combat environment is parsed, extracted, and rasterized to construct a marine combat environment for UAV flight simulation. To address the problem of path planning for low-altitude penetration in complex environments, a photosensitivity-enhanced plant growth algorithm (PEPG) is proposed. Based on the plant growth path planning algorithm (PGPP), the proposed algorithm improves upon the light intensity preprocessing and light intensity calculation methods. Moreover, the kinematic constraints of the UAV, such as the turning angle, are also considered. The planned path that meets the safety flight requirements of the UAV is smoother than that of the original algorithm, and the length is reduced by at least 8.2%. Finally, simulation tests are carried out with three common path planning algorithms, namely, A*, RRT, and GA. The results show that the PEPG algorithm is superior to the other three algorithms in terms of the path length and path quality, and the feasibility and safety of the path are verified via the autonomous tracking flight of a UAV.

## 1. Introduction

With the continuous progress of science and technology and the development of unmanned systems, unmanned aircraft combat is gradually becoming the mainstream form of modern warfare. Low-altitude penetration [1], as a tactical means, is an important feature of future UAV combat. In an increasingly complex combat environment, unmanned aerial vehicles (UAVs) need to reduce flight altitudes; strategically avoid anti-aircraft fire and radar monitoring with the help of mountain ranges, islands, and other bunkers; and use low-altitude penetration to complete reconnaissance or strike missions, enhance the survival ability of aircrafts, and improve the precision of strikes. The ocean is one of the main battlefields for military disputes and operations because of its important position in influencing national resources, economic trade, and territorial security. UAVs can complete low-altitude penetration in marine combat environments and successfully execute missions to guarantee the safety of sea territories. These missions can be conducted via two approaches, namely, the construction of combat environments for electronic navigational charts (ENCs) [2] and UAV path planning [3,4].

ENCs are digitalized thematic maps that record geographic information about sea areas. S-57 ENCs [5] are produced by the official departments of waterways in various countries in accordance with the “Digital Hydrographic Data Transmission Standard” issued by the International Hydrographic Organization (IHO) in 1992 and contain the elements of seafloor topography (bathymetry), navigational obstacles, port facilities, tides, currents, and so on. They are object-oriented electronic maps in a vector format, which is convenient for data exchange and transmission. S-57 ENCs are widely used on platforms such as torpedoes [6], ships [7], and UAVs [8] to carry out tasks such as navigation, surveying and mapping, and search and rescue (SAR) in marine environments.

Path planning refers to the rapid planning of a safe path under UAV kinematic constraints by taking into account factors such as the terrain, weather, and threats in the search space. Common path planning algorithms are generally classified into three categories [3,9]: (i) search-based methods, such as the A* algorithm [10] and D* Lite algorithm [11]; (ii) sampling-based methods, such as the RRT algorithm [12,13] and the probabilistic roadmap algorithm (PRM) [14]; and (iii) bionic intelligent methods, such as particle swarm optimization (PSO) [15], genetic algorithm (GA) [16], ant colony optimization (ACO) [17], and the plant growth path planning algorithm (PGPP) [18]. Although search-based methods can ensure the optimal solution, the quality of the solution is limited by the resolution of the search space [19]. Sampling-based methods have advantages in high-dimensional space, but the planning time is stochastic [20]. Bionic intelligent methods simulate biological characteristics to optimize the solution, which has strong environmental adaptability and high efficiency but also has randomness [21].

In recent years, many academics have been researching the problem of low altitude penetration of UAVs and the bionic intelligent planning approach. Ref. [22] proposed an improved A* algorithm with a bidirectional variable step strategy and minimal RCS tactics to realize fast penetration path planning for stealth UAVs in dynamic radar threat scenarios. Refs. [23,24] proposed an improved PSO algorithm to plan a better breakout flight path by optimizing the adaptive function of the algorithm. The PGPP algorithm [18] utilizes the characteristics of plant growth and draws on the basic physiological mechanisms of plant phototropism, dorsotropism, apical dominance, and branching for algorithm design to realize path planning from the starting point to the target point. The PGPP algorithm for path planning has advantages in reducing the number of waypoints and memory overhead. Zhou et al. [25] combined the flight performance of quadrotor UAVs, proposed a new bio-inspired path planning algorithm and verified the effectiveness of the algorithm through actual low-altitude obstacle avoidance flights. However, the majority of existing algorithms consider only simple scenarios with regular obstacles (circles or rectangles) or do not really consider the kinematic constraints of UAVs. In addition, many current path planning algorithms for concave obstacles are solved by transforming the obstacles into regular convex polygons [26,27]. Nevertheless, in actual nautical charts, the land and the islands are a large number of complex concave obstacles. These methods are inefficient and unable to address the path planning problem properly in complex nautical chart environments.

In this paper, we focus on UAV low-altitude penetration missions in a realistic and complex marine combat simulation environment, which includes numerous irregular obstacles. A new path planning algorithm, the photosensitive-enhanced plant growth algorithm (PEPG), is proposed, which is based on the PGPP algorithm and incorporates UAV turning angle constraints. The main contributions of this paper are summarized as follows:Based on an S-57 electronic chart file and the Mercator projection, information on obstacles such as land islands is extracted, rasterized, and combined with threat information such as radar, and an accurate marine combat simulation environment is constructed.Aiming at a low-altitude penetration mission in the above complex concave obstacle combat environment, a PEPG algorithm based on the PGPP algorithm is proposed, which improves the light intensity preprocessing and other parts, combines with the UAV’s turning angle constraints, enhances the algorithm’s heuristic information in the path search, and the planned path is shorter and smoother.The three common path planning algorithms are compared and contrasted via simulation tests, and the effectiveness of the algorithm proposed in this paper is verified.

## 2. Complex Marine Combat Environment Construction

### 2.1. Construction of the Gridded Nautical Chart Environment

Environmental modeling is the first prerequisite for realizing the whole planning task. In the UAV low-altitude penetration path planning task, the obstacles affecting flight safety are mainly land and islands in the marine environment.

The information in S-57 ENCs is saved and plotted hierarchically [28]. All the elements belong to the corresponding layers according to the type of object, and the geometric information of the objects needs to be extracted from the corresponding layers. The vector data model adopted in S-57 ENCs describes real-world objects through feature objects and spatial objects [29], and each object is a combination of feature objects and spatial objects. The feature object label contains the description information of the object, and the spatial object label contains the geometric information of the object, which contains the spatial position of the object in terms of points, lines and surfaces.

To accurately obtain the land and island information of S-57 ENCs, it is necessary to analyze and read the S-57 ENC file data on the basis of the theoretical model and the data structure of the above nautical chart data. The process of land information extraction is shown in Figure 1, which ultimately yields a set of latitudinal and longitudinal coordinates of the land and islands characterized by polygonal boundaries.

To reflect the real shape of the land islands in the nautical charts and ensure the accuracy of navigation, the Mercator projection [30] is adopted in this paper to convert the latitudinal and longitudinal coordinates into plane Cartesian coordinates to ensure that the shape of the region remains unchanged. The equator is selected as the reference latitude line, the prime meridian is the central meridian, the intersection of the two is the coordinate origin, the east-to-north direction is the positive direction, the earth is regarded as a sphere, and the chart is mapped in the plane with the following formula:(1)$$\left.\left\{\begin{array}{cc} x & =R_{0}\lambda \\ y & =R_{0}\ln\left(\tan\left(\frac{\pi }{4}+\frac{\phi }{2}\right)\right) \end{array}\right.\right.$$
where λ denotes the geographic longitude (radian system), ϕ denotes the geographic latitude (radian system), and the equatorial radius is $R_{0}=6,378,137$ m.

As shown in Figure 2a, an electronic chart of a sea area is selected as an example. The latitudinal and longitudinal ranges of the sea area are 116.500° E to 123.450° E and 31.500° N to 39.28° N, and the Mercator projection is carried out on the land islands represented by the latitudinal and longitudinal coordinates $\left(\phi ,\lambda \right)$ in the electronic chart to obtain the chart information corresponding to the Mercator plane coordinates $\left(x,y\right)$. As shown in Figure 2b, the transformed land islands are complex planar polygons after the Mercator projection.

To accurately describe and effectively characterize the obstacle information in the chart environment, the raster method is adopted in this paper to further model the chart environment information. First, we perform binarization on the chart after Mercator projection and set the obstacle-free area as white and the obstacle area as black, as shown in Figure 2c. Then, the appropriate raster search space size is selected according to the latitudinal and longitudinal boundary lengths of the nautical chart, and the land and island information in the nautical chart is rasterized to obtain the raster coordinates of each region. In the case of a certain chart size, the smaller the size of the raster unit is, the greater the degree of search space refinement, and the greater the amount of computation.

In this paper, a 100×100 search space is selected for the construction of a complex environment, and the simulation environment model effect is shown in Figure 2d.

### 2.2. Gridding of the Threat Information

When UAVs are used to perform combat missions, in addition to land and islands at sea, there are threats such as adverse weather conditions and enemy radar and fire. The radar threat is determined by the radar position and radar detection range; the fire threat is determined by the fire point and fire kill radius. In this paper, the threats are modeled uniformly with the following formulas:(2)$${\left(x-x_{j}\right)}^{2}+{\left(y-y_{j}\right)}^{2}\leq r^{2}.$$
where $\left(x_{i},y_{i}\right)$ and *r* are the center point location and threat radius, respectively, of each type of threat.

Weather, radar, and fire threat information is combined with land island terrain information in the chart, and finally, the complex marine environment is modeled. The results are shown in Figure 3.

Finally, all the obstacles in the search space occupied by the raster unit are saved in the form of a collection, which is labeled $OBS=\left\{O_{1},O_{2}\ldots O_{n}\right\}$. The remaining raster unit is the feasible area. Moreover, the grid environment information of the complex nautical chart is obtained and is used as the known information for subsequent path planning.

## 3. Photosensitivity-Enhanced Plant Growth Algorithm

Based on the known information of the above environment and UAV turning angle constraints, the photosensitivity-enhanced plant growth algorithm (PEPG) for UAV path planning in complex concave obstacle environments based on the PGPP algorithm is proposed in this paper, which enhances the algorithm’s accurate perception of light sources during bud growth.

### 3.1. Fundamentals of the PGPP Algorithm

The basic principle of the PGPP algorithm [18] is shown in Figure 4, where the starting point of the flight path planning is assumed to be a plant seed germ, the end point is assumed to be the light source required for plant growth, and the shaded and unshaded areas of the search space are obtained by preprocessing the environment with regards to the light intensity. The algorithm discretizes the growth process of the plant from the seed germ to the light source in a growth cycle first according to the bud age of each bud to determine whether to carry out random branching to produce new buds. Then, the current buds grow in the direction of the growth vector. Once the current buds reach the light source, the algorithm ends, and the growth path is obtained.

### 3.2. Preprocessing of the Light Intensity Values in Complex Environments

Light intensity value preprocessing is used to provide heuristic information for bud growth by simulating the natural environment and projecting light from a light source to the entire search space. The search space can be divided into the following three states: the obstacle area, unshaded area and shaded area. The light intensity value of each raster cell is determined by a combination of its distance from the end point, whether it is an obstacle and whether it is shaded by an obstacle. The calculation formula is as follows:(3)$$\left.\left\{\begin{array}{cc} & L\left[x\right]\left[y\right]=K\times \frac{L_{init}\left[x\right]\left[y\right]}{\sqrt{l}}\\ & l=\sqrt{{\left(x-x_{goal}\right)}^{2}+{\left(y-y_{goal}\right)}^{2}} \end{array}\right.\right..$$
where the initial light intensity $L_{init}\left[x\right]\left[y\right]$ of all grid cells $\left(x,y\right)$ in the search space is set to 150. Moreover, the light intensity coefficient *K* is 1 when the grid is represented as unobstructed, 0.4 when it is obstructed, and 0 when it is represented as an obstacle. $\left(x_{goal},y_{goal}\right)$ is the coordinate of the grid cell at the end point. Preprocessing of the light intensity values in the simulation environment is the core step of the algorithm and provides key heuristic information for the growth of buds out of the path toward the end point and timely obstacle avoidance. The method of light intensity preprocessing in the PGPP algorithm [18] is shown in Figure 5, and in previous studies, two convex obstacles, rectangular and circular, were mainly considered. In particular, all obstacles were computed one by one throughout the entire search space; the vertices of rectangular obstacles were connected sequentially to confirm the two tangents with the largest tensor angles; and the geometrically symmetric centers of circular obstacles were connected to the end points to determine the end point and the tangent point of the obstacle. The area that satisfies the requirement of being within the tangent line and not between the obstacle and the end point is considered to indicate occlusion. This method of using the tangent point to determine the calculation of the shaded area in the search space can be applied only to an environment with a small number of regular geometric obstacles and is not applicable to an environment with complex geometric grid obstacles, which restricts the application of the PGPP algorithm.

To obtain important occlusion region information for light intensity preprocessing in a complex concave obstacle rasterized navigation environment, we implement the Bresenham straight-line algorithm [31] to determine the light intensity coefficients of the raster cells in the search space.

The steps of the algorithm are as follows:

Step 1: Set the light intensity coefficient *K* of the obstacle grid in the search space to 0 and initialize all the remaining grids to be in the shaded state, which is set to 0.4.

Step 2: Calculate the rasters of the unobstructed region in the search space. Iterate over the boundary rasters of the entire search space and note that the coordinates of the current boundary raster cell are $\left(x_{m},y_{m}\right)$.

Step 3: Connect a line from the light source (that is, the end point $\left(x_{g},y_{g}\right)$) to $\left(x_{m},y_{m}\right)$, calculate the absolute value of the slope |k| of the straight line, and initialize the cumulative error variable *e* to 0. Starting from the end point $\left(x_{g},y_{g}\right)$, the coordinate of the raster cell $\left(x_{i},y_{i}\right)$ is iterated along the straight line to the coordinate of the boundary raster cell $\left(x_{m},y_{m}\right)$, and the iteration directions are $x_{dir}=\left(x_{m}-x_{g}\right)/\left|x_{m}-x_{g}\right|$ and $y_{dir}=\left(y_{m}-y_{g}\right)/\left|y_{m}-y_{g}\right|$. If |k|≥1, then proceed to Step 4; otherwise, skip to Step 5.

Step 4: Add 1 to *e* before each grid cell is checked during iteration. If e≥|k|/2, then update the current grid cell coordinates to $\left(x_{i}+x_{dir},y_{i}+y_{dir}\right)$. At the same time, subtract |s| from *e*. Otherwise, update the coordinates to $\left(x_{i},y_{i}+y_{dir}\right)$. Determine whether the coordinates of the obstacle grid are updated. If they are, then the current round of iteration is completed, and Step 6 can be implemented. Otherwise, set the current grid cell light intensity coefficient *K* to 1, and continue to update the grid cell coordinates until the current boundary grid cell is $\left(x_{m},y_{m}\right)$.

Step 5: Add |k| to *e* before each grid cell is checked during the iteration. If e≥1/2, then update the current grid cell coordinates to $\left(x_{i}+x_{dir},y_{i}+y_{dir}\right)$. At the same time, subtract 1 from *e*. Otherwise, update the coordinates to $\left(x_{i}+x_{dir},y_{i}\right)$. Next, determine whether the coordinates of the obstacle grid are updated. If they are, then, after the curret round of iteration is completed, proceed to Step 6. Otherwise, set the current grid cell light intensity coefficient *K* to 1 and continue to update the grid cell coordinates until the current boundary grid cell is $\left(x_{m},y_{m}\right)$.

Step 6: Determine whether all boundary grid cells in the search space have been traversed. If so, the calculation of the light intensity coefficient in the occlusion area is completed. Otherwise, return to Step 2.

For the light intensity preprocessing of complex environment charts, the final calculation of the shaded area is shown in Figure 6. The light intensity preprocessing method of this paper’s algorithm can fully perceive environmental information and provide more effective light intensity-inspired information for subsequent bud growth.

### 3.3. Sector Light Intensity Vector Homogenization

The calculation of the light intensity vector is a key step in bud growth updating, and the next light intensity vector $G_{L}$ relies on calculating the light intensity values in the reference subregion. The light intensity vector is calculated by taking the line between the current bud and the end point as the axis of symmetry, selecting the semicircle facing the end point as the reference area, dividing the reference area into twelve sectors, summing the light intensity values of the grid cells in the sub-areas, sorting the sub-areas according to the size of the light intensity values, selecting the bisector of the sector with the largest light intensity as the direction of the light intensity vector, and summing the light intensity values of the grid nodes in the sector as the size of the light intensity vector. Since the raster map is a discrete space, the number of raster cells contained in sectors of the same area may be different, as shown in Figure 7. For example, with a radius of the reference area of 5, the number of raster cells in different sectors is 2∼4, and the optimal direction of the current bud growth cannot be truly reflected simply by the cumulative method.

Therefore, the calculation method for the light intensity vector direction is improved, the light intensity $L_{sector}$ of each sector is averaged according to the number of raster cells, the calculation formula is shown in Equation (Equation 4), and the sub-areas are sorted according to the average value $L_{sector}$.
(4)$$L_{sector}=\frac{1}{m}\sum_{i=1}^{m}L_{i}$$

Here, *m* is the number of grids in each sector, and $L_{i}$ is the corresponding light intensity value of each grid. The selection of the radius of the reference area also affects the exploration range of the current buds, and the size should be selected appropriately. If the radius is too large, the end point will not be searched when the end point is located in the sector that contains obstacles, which will lead to a failure in the path planning. If the radius is too small, it will not be able to sense the obstacles in front of it in time, which will increase the risk of failure of the planning. In this paper, the search space is set to 100×100, the range of the reference radius is reasonably adjusted according to the distance between the current bud and the end point, and the formula is as follows:(5)$$r=\max\left(\min\left(\sqrt{{\left(x_{p}-x_{goal}\right)}^{2}+{\left(y_{p}-y_{goal}\right)}^{2}},8\right),5\right)$$
where $\left(x_{p},y_{p}\right)$ is the grid cell coordinate of the current bud, and $\left(x_{g}oal,y_{g}oal\right)$ is the grid cell coordinate of the end point.

### 3.4. Growth Vector Calculations Considering Turning Angle Constraints

The growth vector of a plant is affected by the light intensity vector, the gravity vector, and the growth vector of the previous cycle. The calculation formula is as follows:(6)$$\left\{\begin{array}{c} G_{\mathrm{New}\;}=K_{n1}\cdot G_{L}+K_{n2}\cdot G_{R}+K_{n3}\cdot G_{\mathrm{Last}\;}\\ G_{L}/∥G_{L}∥=I_{\max\;}\\ ∥G_{L}∥=L_{\mathrm{sector}}^{\max}\\ G_{R}=\left(x-x_{\mathrm{start}}\right)i+\left(y-y_{\mathrm{start}}\right)j \end{array}\right.$$
where $G_{\mathrm{New}}$ is the new growth vector; $G_{L}$, $G_{R}$, and $G_{\mathrm{Last}}$ are the light intensity vector, the gravity vector, and the last cycle growth vector, respectively; $K_{n1}$, $K_{n2}$, and $K_{n3}$ are the corresponding weighting coefficients; $I_{\max}$ is the unit vector in the direction of the current maximum light intensity of the bud; $L_{\mathrm{sector}}^{\max}$ is the light intensity value of the corresponding sector; and $\left(x_{start},y_{start}\right)$ is the coordinates of the grid cell at the starting point.

For UAVs, the turning angle is limited by the maximum overload, and to ensure the feasibility of the path, a turning angle constraint is added during the path point planning process. When the turning angle does not satisfy the constraint, the next direction with a larger light intensity value is selected in order of the light intensity value in the reference area.

As shown in Figure 8, $G_{\mathrm{Last}}$ is the growth vector of the current bud in the last cycle; $\varphi_{\max}$ is the maximum turning angle; and the numbers on the arcs represent the order of the light intensity values of the sectors. Thus, the current bud growth will be carried out in the feasible region within the range $\left(-\varphi_{\max},\varphi_{\max}\right)$ (green part in the figure). Since the growth vectors $G_{1}$ and $G_{2}$ synthesized from the light intensity vectors corresponding to Sectors 1 and 2 with the largest light intensity values are discarded because they do not satisfy the turning angle constraint, the current bud selects the light intensity vectors corresponding to Sector 3 to synthesize the growth vectors $G_{3}$, which satisfy the turning angle constraint and are used for the growth of the bud in the current cycle.

The pseudocode of the PEPG algorithm is shown in Algorithm  1.
**Algorithm 1:** PEPG
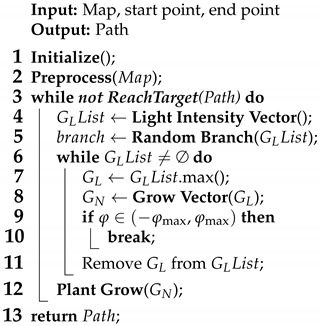


The start and end points are set as the germ and the light source, respectively, and the light intensity value of the grid node is calculated according to the blockage of the light source by the grid obstacle. The light intensity vector of the current bud was selected according to the light intensity value in the selected sector, and the current cycle growth vector was calculated by combining the gravity vector and the previous cycle growth vector. The bud grew along the direction of the growth vector that met the constraints of the turning angle and was collision-free.

When the branches grow to a certain age, a random method is used to determine whether they can be branched. During branching, a direction with greater light intensity is randomly selected according to the light intensity distribution of the original bud for calculation and storage of the side bud. When the main bud is withered, the side bud is removed, and the plant is allowed to grow until the light source is found.

### 3.5. Planning Effect of the PEPG Algorithm

To verify the effectiveness of the algorithm proposed in this paper, it is compared with the PGPP algorithm in terms of its effect on path planning, as shown in Figure 9. To reduce the influence of the search space and start and end point selection on the algorithm planning results, 10 groups of different start and end points are selected, and the simulation data are shown in Table 1, where the coordinates are the raster coordinates after latitude and longitude transformations and the length of the path is the length of the search space.

As shown in Figure 9, because the PGPP algorithm [18] treats obstacles as two kinds of convex obstacles, circular and rectangular, and cannot handle the information of concave obstacles well in the search space, part of the flyable area in the search space cannot be used for path planning, and the light intensity illumination information is insufficient, resulting in the planned path failing to cross the strait and increasing the path length. However, the PEPG algorithm can make effective use of the obstacle information in the search space. Because the PGPP algorithm does not consider turning angle constraints, there are large turning angles in the channel, and the algorithm does not effectively utilize light intensity values at the early stage of path growth due to the lack of sector light intensity value homogenization, resulting in a zigzag path. The PEPG algorithm is planned in a grid obstacle environment with a clear growth direction and a smoother path, and the length of the planned path is at least 8.2% shorter than that of the PGPP algorithm.

## 4. Simulation Test

### 4.1. Algorithm Simulation and Comparison Verification

In this paper, Python is used to carry out simulation experiments which are run on a computer with a 3.70 GHz CPU and 32 GB of RAM. In two scenarios, the most representative algorithms of the three path planning algorithms, namely, the A* algorithm, RRT algorithm, and GA, are selected for comparison with the PEPG algorithm, and the performance of the algorithms is analyzed based on the algorithm’s three indices of the planning time, number of planned waypoints, and path length. To reflect the differences in the performances of different algorithms and reduce the impact of algorithmic randomness in the single planning process, the index data are selected as the average of 30 runs of the four algorithms.

#### 4.1.1. Complex Scenarios without Threats

To test the planning ability of the four algorithms for environments with concave obstacles, a non-threatening complex scenario (referred to as “Scenario 1”) was used to select part of the sea area in the C110408A.000 chart as the search space, and the selected sea area ranged from 106.266° E to 114.365° E and 17.010° N to 24.580° N. With (107.231° E, 20.116° N) as the starting point and (113.484° E, 18.250° N) as the end point, the planning effect is shown in Figure 10 and Figure 11, and the algorithm performance comparison data are shown in Table 2.

In terms of the path length, both the A* algorithm and the PEPG algorithm are able to stabilize the path length at a small value, while the GA and RRT algorithms are longer and have more ups and downs. The RRT algorithm can find feasible solutions and has the longest path length, with an average path length of 116.09. The GA has a shorter path length than the RRT algorithm. The GA has an evolutionary mechanism, and the use of cross-mutations and other characteristics can optimize the path. In the case of sufficient planning time, the path can approach the optimal solution, but easily fall into the local extremes, preventing the optimal solution from being found. The average path length planned by A* algorithm is 89.11, and the characteristic of optimal resolution makes the path planned by the algorithm very short. However, because only the grid-by-grid operation can be planned, the path length is not the actual optimal value. The PEPG algorithm is not restricted by neighboring rasters when planning waypoints and has a shorter planned path length than the A* algorithm in Scenario 1, with an average path length of 87.02.

In terms of planning time, the A* algorithm and the PEPG algorithm are faster than the RRT algorithm and the GA in most cases, as shown in Figure 11b. The RRT algorithm performs waypoint planning by random sampling, and the time needed to find the end point is uncertain. Moreover, the GA generates the initial solution through the initial population and then iteratively converges the waypoints through cross-variation on the waypoints to find the waypoints. Additionally, randomness occurs, and both algorithms have longer average planning times.

The PEPG algorithm has a slightly slower planning speed than does the A* algorithm. However, the path length is shorter, the waypoints are smoother, and the number of waypoints is much less than that of the A* algorithm. This is still because the A* algorithm can only plan raster cell by raster cell and does not consider the turn cost and turning angle constraints, which leads to many planned waypoints and zigzagging paths. These paths may not necessarily be able to be used as feasible paths for UAVs during tracking flights and must be further processed using path optimization methods. In contrast, the step size of the PEPG algorithm can be used to calculate the growth of new shoots based on the growth vector without the need to plan one by one raster cells, which can generate fewer waypoints. Additionally, the turning angle constraints are taken into account during the planning process, and the planned waypoints satisfy the turning angle constraints, which can be used for direct flight tracking.

#### 4.1.2. Complex Scenarios with Threats

To further test the planning ability of the algorithm in complex combat environments, weather, enemy radar and fire threats are added on the basis of concave obstacles, and part of the sea area in the C110408A.000 chart is selected as the simulation test area of the complex scenario with threats (referred to as “Scenario 2”). The range of the selected sea area is as follows: 116.500° E to 123.450° E and 31.500° N to 39.407° N. the threat information is shown in Table 3. Taking (119.466° E, 39.154° N) as the starting point and (123.109° E, 33.386° N) as the end point, the planning effect of the four algorithms is shown in Figure 12, and the results of the algorithm comparison data are shown in Table 4.

The experimental results of Scenario 2 show that all four path planning algorithms can guarantee finding the path. Compared with Scenario 1, the straight-line distance between the start and end points in Scenario 2 is slightly longer, and the environment becomes more dense and complex from empty to sparse. The RRT algorithm needs to sample more times in the narrow space, which consumes more planning time, and the average planning time of the RRT algorithm becomes the longest. The average planning time of the GA is relatively stable compared with that of Scenario 1, indicating that the planning of the GA is less affected by the complex environment. The average planning time of the A* algorithm is increased because the planning time relies on the number of searching raster cells during the expansion of waypoints. A longer distance between the start and the end increases the computational cost of the A* algorithm. Additionally, the calculation still cannot take into account the cost of the turn, and there are more turning points for the planned waypoints. In terms of the path length and planning effect, the PEPG algorithm still plans the shortest and smoothest waypoints among the four algorithms.

### 4.2. Tracking Control Test

To further verify the feasibility of the waypoints, a PID method is adopted to control the UAV to track the waypoints and realize a low-altitude penetration mission. The position of the UAV is converted from Geodetic Coordinate System to North-East-Down Coordinate System. The axial overload, normal overload and roll angle information obtained by controller solving are jointly inputted into the UAV dynamics model, which solves the position, attitude, and velocity information of the UAV, and the results are fed back to the controller to track the UAV’s waypoints.

#### 4.2.1. UAV Dynamics Model

When the UAV is tracked and controlled according to the planned waypoints, the UAV dynamics model must determine the flight state at each instant. Assuming that there is no sideslip in flight and that the thrust is along the velocity direction, the dynamics model is divided into UAV kinematics equations and UAV dynamics equations. The UAV kinematics equations used in this paper are as follows:(7)$$\left\{\begin{array}{c} {\dot{x}}_{m}=V_{m}\cos\gamma_{m}\cos\chi_{m}\\ {\dot{y}}_{m}=V_{m}\cos\gamma_{m}\sin\chi_{m}\\ {\dot{z}}_{m}=-V_{m}\sin\gamma_{m} \end{array}\right.$$
where $\left(x_{m},y_{m},z_{m}\right)$, $V_{m}$, $\chi_{m}$ and $\gamma_{m}$ denote the position coordinates of the UAV, the rate, the heading angle, and the climbing angle, respectively.

The UAV dynamics equations are as follows:(8)$$\left\{\begin{array}{c} {\dot{V}}_{m}=g\left(n_{mx}-\sin\gamma_{m}\right)\\ {\dot{\chi }}_{m}=gn_{mn}\sin\mu_{m}/V_{m}\cos\gamma_{m}\\ {\dot{\gamma }}_{m}=g\left(n_{mn}\cos\mu_{m}-\cos\gamma_{m}\right)/V_{m} \end{array}\right.$$
where the axial overload $n_{mx}\in \left\lceil -2,2\right\rceil$, normal overload $n_{mn}\in \left\lceil -2,2\right\rceil$, roll angle $\mu_{m}\in \left\lceil -2,2\right\rceil$, and *g* is the gravitational acceleration.

#### 4.2.2. Control Law Design

The UAV tracks the waypoints with control, and to solve the required control quantity $\left[n_{mx},n_{mn},\mu_{m}\right]$, the control law is designed as follows:(9)$$\left\{\begin{array}{c} n_{mx}=K_{P}^{V}\left(V_{r}-V\right)+K_{D}^{V}\frac{d\left(V_{r}-V\right)}{dt}\\ n_{mn}=K_{P}^{\gamma }\left(\gamma_{r}-\gamma \right)+K_{D}^{\gamma }\frac{d\left(\gamma_{r}-\gamma \right)}{dt}\\ \mu_{m}=K_{P}^{\chi }\left(\chi_{r}-\chi \right)+K_{D}^{\chi }\frac{d\left(\chi_{r}-\chi \right)}{dt} \end{array}\right.$$
where $K_{P}^{V}$, $K_{P}^{\gamma }$, and $K_{P}^{\chi }$ are the coefficients of the proportional term; $K_{D}^{V}$, $K_{D}^{\gamma }$, and $K_{D}^{\chi }$ are the coefficients of the differential term; the desired speed $V_{r}$ is a fixed value; and the desired climbing angle $\gamma_{r}$ and heading angle $\chi_{r}$ are determined by the difference between the next waypoint and the current position. The overall block diagram of path control tracking is shown in Figure 13.

#### 4.2.3. Tracking Control Effect

The planning results of the four algorithms in Scenario 1 are subjected to UAV waypoint tracking control simulation tests, and the results are shown in Figure 14.

As shown in the figure above, when the waypoints planned by the four algorithms are utilized directly for UAV control tracking, the RRT algorithm fails to track at the 9th waypoint due to unfulfilled turning angle constraints. The other three algorithms can realize all the waypoints on the tracking, but the path planned by the GA and the A* algorithm is tortuous. Furthermore, due to the A* algorithm’s large turning angle at the corner, the UAV repeatedly appears to re-search and track the previous waypoint, resulting in large maneuvers, which not only reduces flight efficiency but also increases the risk that the UAV will be detected by the radar and other sensors. Compared with those of the other three algorithms, the waypoints planned by the PEPG algorithm can enable the UAV to accurately follow all the waypoints within a smaller maneuvering range.

The tracking control test verifies that the paths planned by the PEPG algorithm are smooth and can be directly applied to control and tracking without postprocessing, which is expected to reduce the detectability of UAVs in low-altitude penetration while improving navigation efficiency and further improving the safety and stealth of UAV flight.

## 5. Conclusions

To solve the low-altitude penetration problem of UAVs in complex marine combat environments, obstacle information from islands is first extracted in this paper based on electronic chart information in the S-57 format. Moreover, atmospheric conditions, radar monitoring and fire threats are comprehensively considered to construct a model for marine combat simulation environment rasterization. Then, based on the PGPP algorithm, a PEPG algorithm is proposed to enhance the algorithm’s accurate perception of light sources when the bud grows. The following conclusions are obtained: ➀ The proposed algorithm is combined with the Bresenham linear algorithm to calculate the light intensity coefficient of the rasterized area so that the algorithm in this paper can be adapted to the rasterized environment of complex concave obstacles and enhance the heuristic information of bud growth when avoiding obstacles and growing toward the end point. ➁ By improving the light intensity vector computation, the sector light intensity values are homogenized, and turning angle constraints are added to ensure that the UAV maneuverability is met. Finally, a simulation test is designed as an example of a low-altitude penetration mission in real combat scenarios to verify the effectiveness of the proposed algorithm in low-altitude penetration combat scenarios in marine environments. The test results reveal that the path length of the proposed algorithm in both scenarios is greater than that of the other three comparative algorithms, and there are fewer waypoints and smoother paths, which meets the maneuverability of the UAV. Moreover, tracking control via the three-degree-of-freedom model verifies the validity and safety of the planned waypoints.

The research problem in this paper is the single UAV low-altitude penetration problem when all the threats are known. Moreover, the PEPG algorithm adapts well to complex environments, and its advantages are better reflected in local unknown environments. There is still potential for the PEPG algorithm to accelerate the computing time. In the future, we will further consider practical application scenarios and study the multi-aircraft low-altitude cooperative penetration problem under the scenario of uncertain threat information.

## Figures and Tables

**Figure 1 biomimetics-09-00212-f001:**
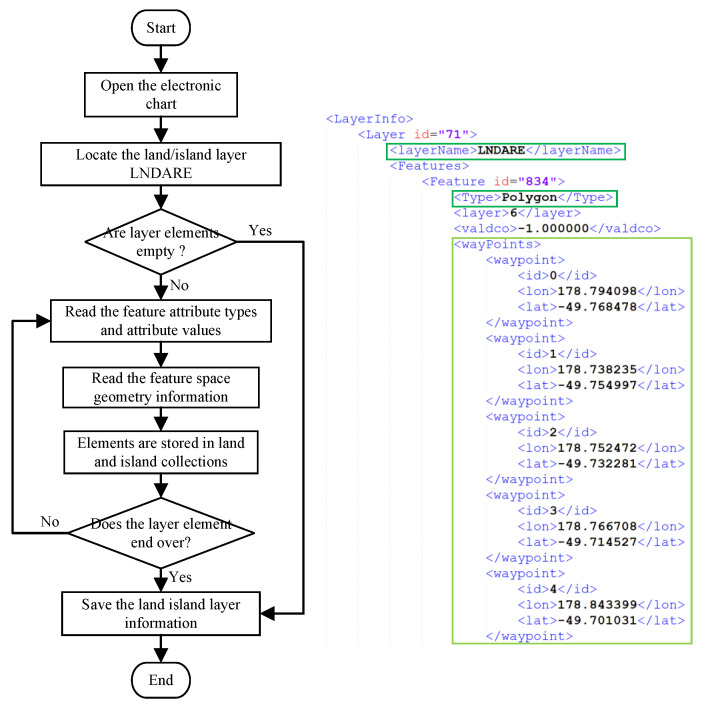
Land information extraction flow chart and results.

**Figure 2 biomimetics-09-00212-f002:**
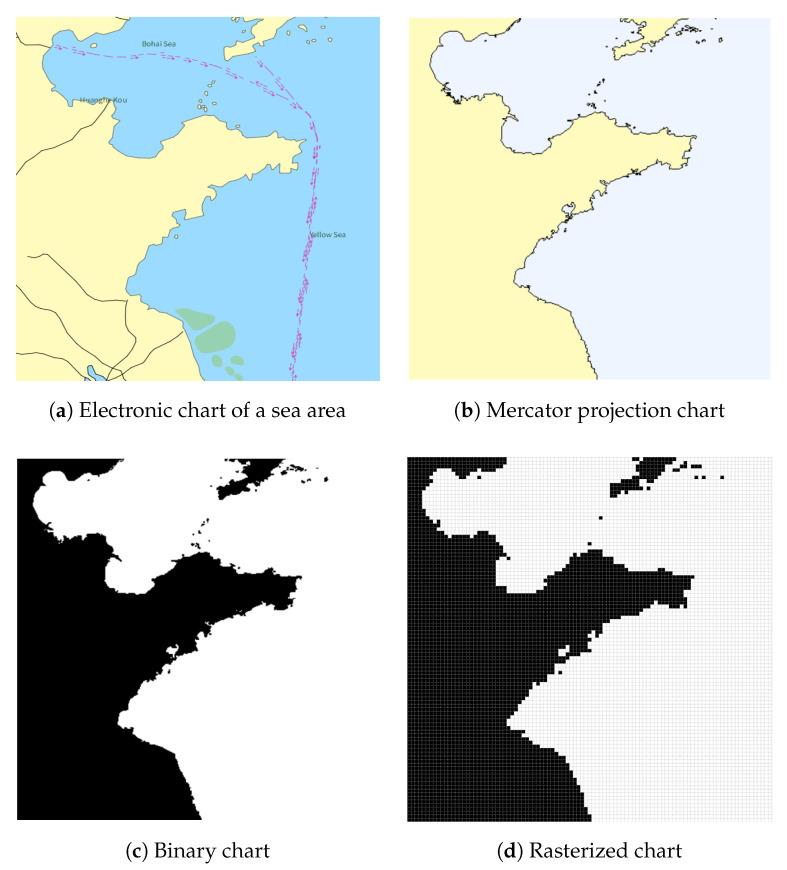
Rasterization of the S-57 EC environment.

**Figure 3 biomimetics-09-00212-f003:**
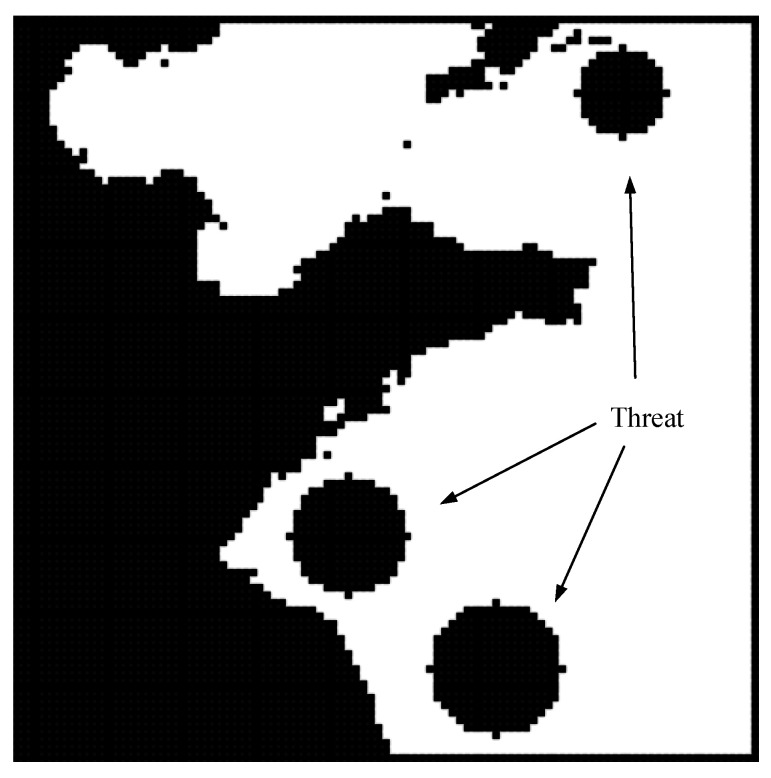
Scenario of complex ocean environments.

**Figure 4 biomimetics-09-00212-f004:**
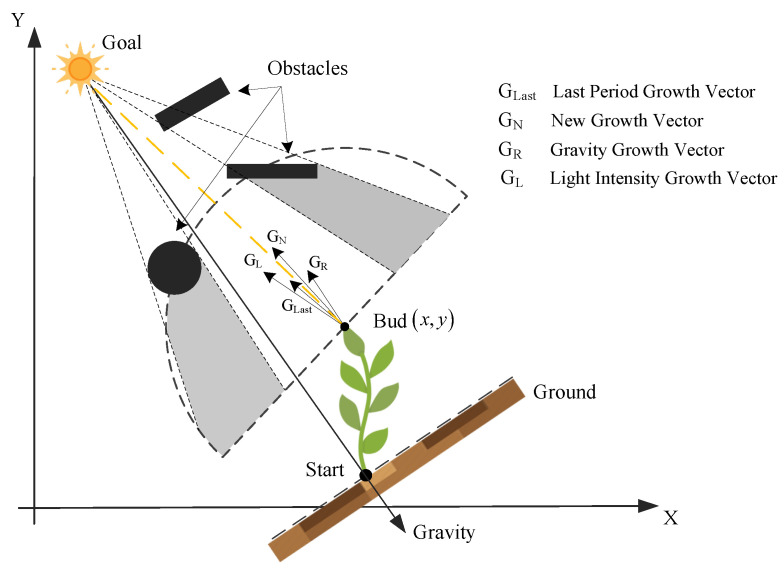
Schematic diagram of the PGPP.

**Figure 5 biomimetics-09-00212-f005:**
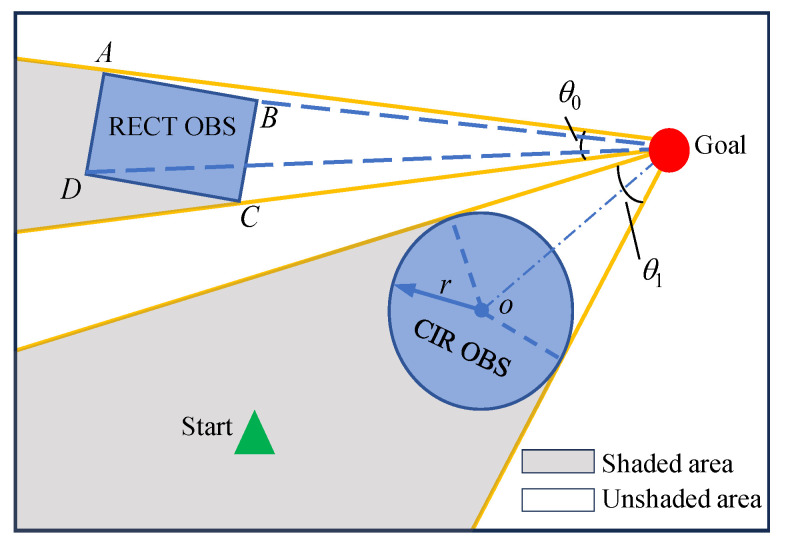
Light intensity preprocessing method for PGPP.

**Figure 6 biomimetics-09-00212-f006:**
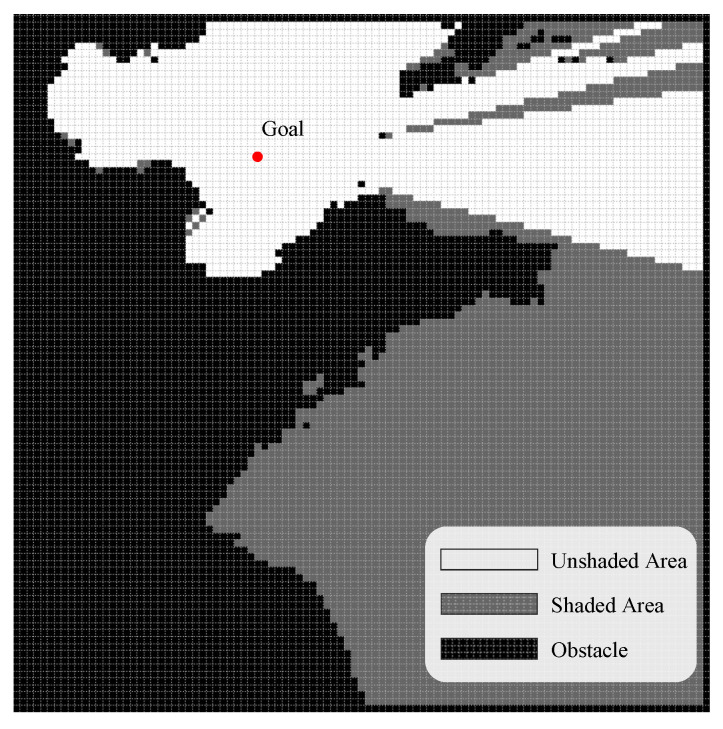
Light intensity preprocessing method for PEPG.

**Figure 7 biomimetics-09-00212-f007:**
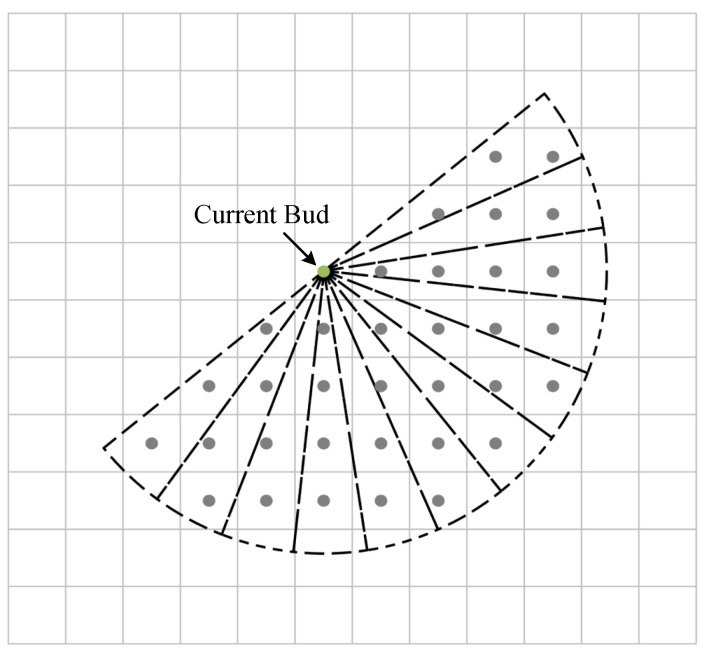
Calculation of the sector light intensity.

**Figure 8 biomimetics-09-00212-f008:**
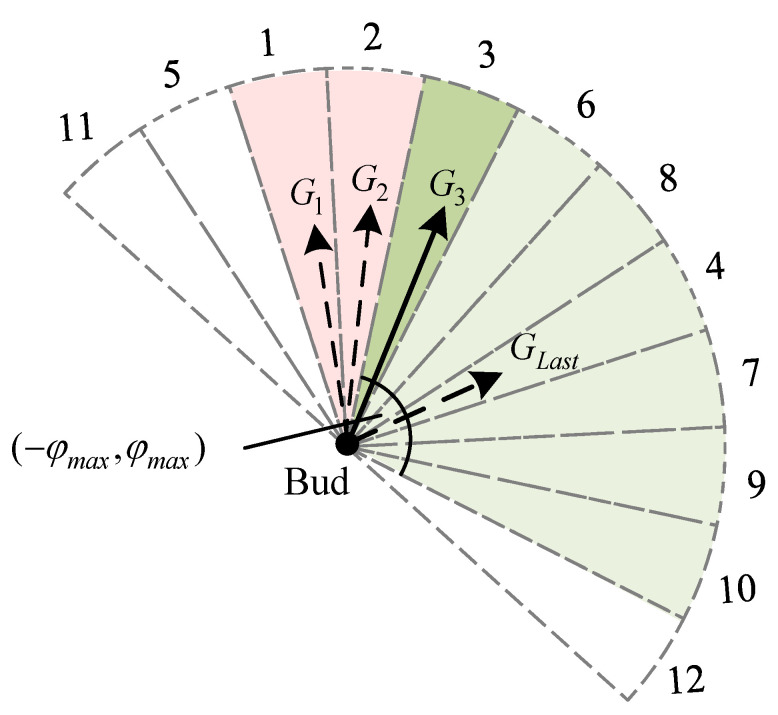
Selection of new bud.

**Figure 9 biomimetics-09-00212-f009:**
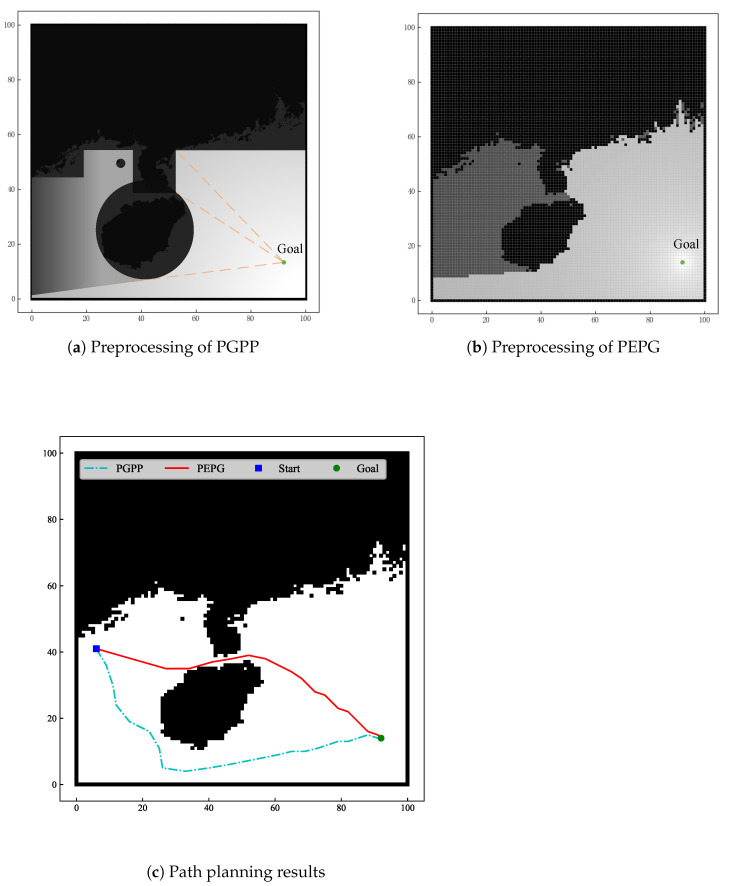
Path planning results of the two algorithms.

**Figure 10 biomimetics-09-00212-f010:**
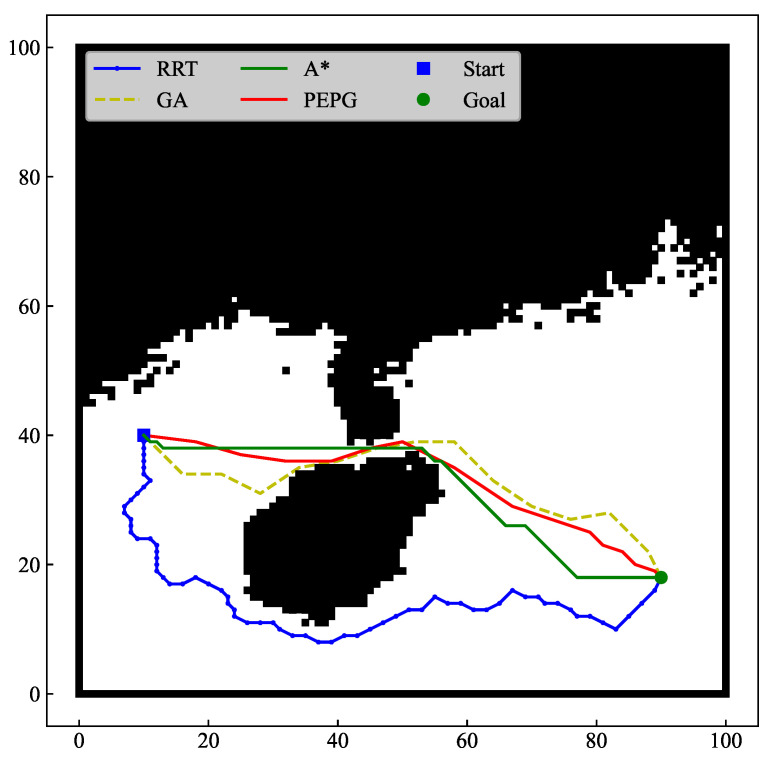
Path planning results of the four algorithms in Scenario 1.

**Figure 11 biomimetics-09-00212-f011:**
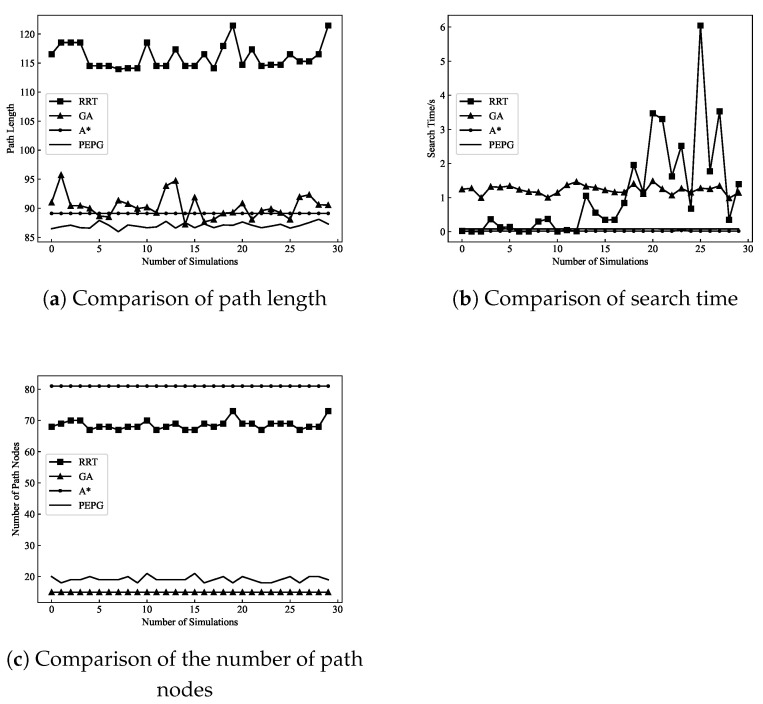
Simulation data of four algorithms in Scenario 1.

**Figure 12 biomimetics-09-00212-f012:**
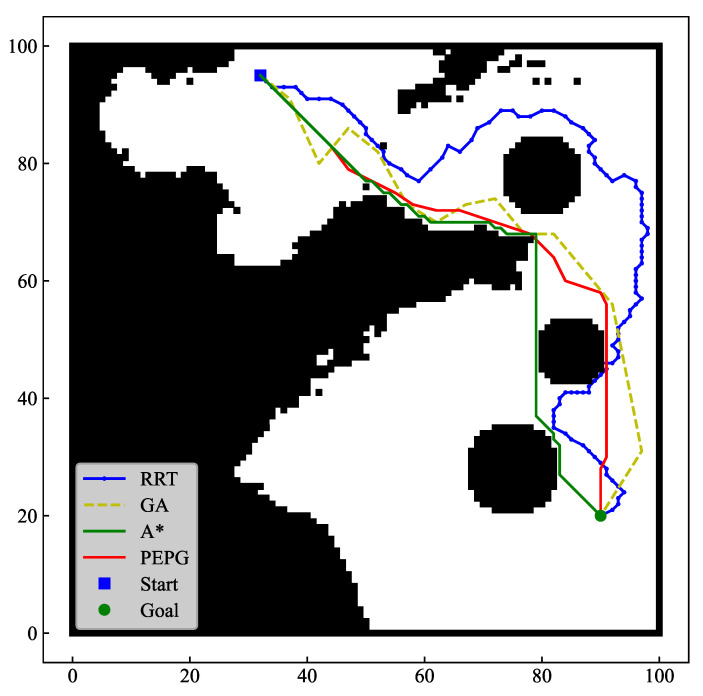
Path planning results of the four algorithms in Scenario 2.

**Figure 13 biomimetics-09-00212-f013:**
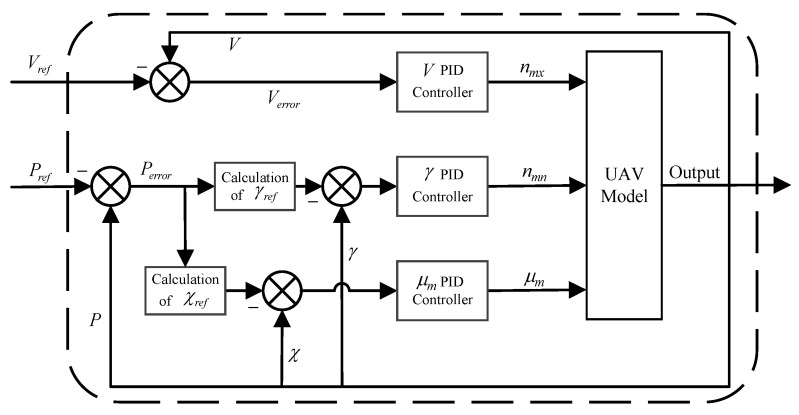
Diagram of the tracking control.

**Figure 14 biomimetics-09-00212-f014:**
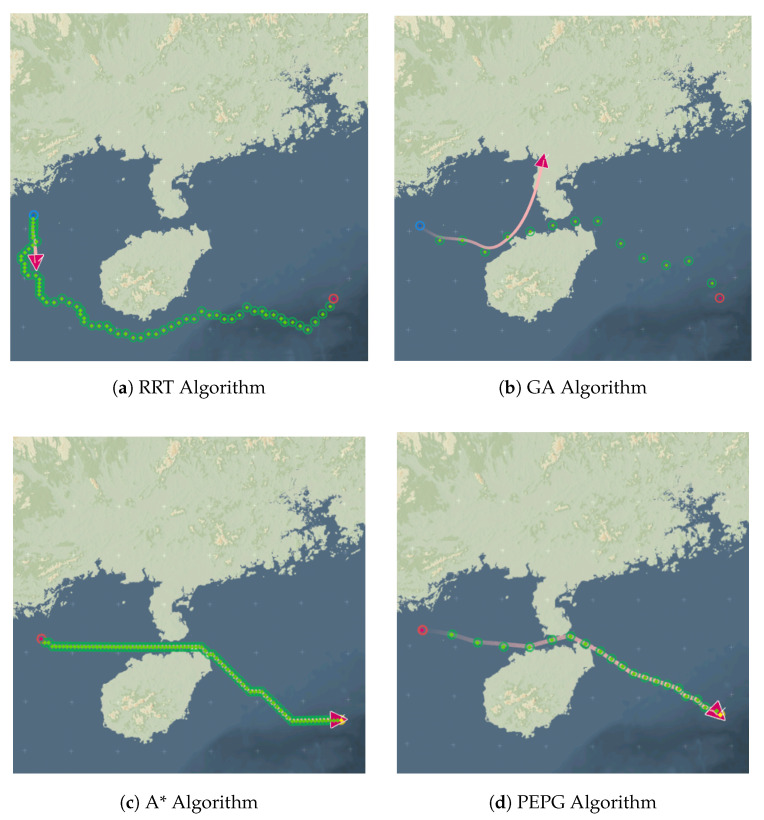
Tracking control renderings of four algorithms.

**Table 1 biomimetics-09-00212-t001:** Comparison of two algorithms simulation results.

No.	Start and Goal Node	PGPP Path Length	PEPG Path Length
1	(12, 36), (96, 42)	105.61	85.51
2	(5, 25), (94, 20)	121.01	102.33
3	(6, 41), (92, 14)	111.58	96.65
4	(5,10), (90, 53)	108.42	99.52
5	(10, 20), (85, 40)	99.34	84.04
6	(85, 40), (10, 20)	101.43	92.49
7	(90, 53), (5,10)	123.10	104.27
8	(92, 14), (6, 41)	114.38	99.78
9	(94, 20), (5, 25)	117.18	104.99
10	(96, 42), (12, 36)	106.32	85.77

**Table 2 biomimetics-09-00212-t002:** Comparison of simulation results in Scenario 1.

Algorithms	Average Time/ms	Average Path Length	Average Path Nodes
RRT	1078	116.09	68.6
GA	1230	90.30	15
A*	19	89.11	81
PEPG	82	87.02	19.2

**Table 3 biomimetics-09-00212-t003:** Threat information.

Threat	Location	Radius/n Mile
Weather Threat	(122.125° E, 33.914° N)	35
Radar Threat	(122.460° E, 37.860° N)	30
Fire Threat	(122.834° E, 35.495° N)	25

**Table 4 biomimetics-09-00212-t004:** Comparison of the simulation results of the four algorithms.

Algorithms	Mean Time/ms	Mean Path Length	Mean Path Nodes
RRT	2085	162.68	111.43
GA	1853	116.97	15
A*	101	110.74	96
PEPG	341	110.21	37.8

## Data Availability

The raw data supporting the conclusions of this article will be made available by the authors on request.

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
