# Peer review of "A Photosensitivity-Enhanced Plant Growth Algorithm for UAV Path Planning"

_biomimetics, 2024, doi:10.3390/biomimetics9040212_

Round 1
Reviewer 1 Report
Comments and Suggestions for Authors
Strong aspects:
The paper presents the algorithm PEPG which represents an improved version of plant path planning algorithm that is described using biological elements. The proposed PEPG shows slightly better performance in path planning with application to UAV control.
Weak aspects:
The results of the proposed algorithm in the scenario with threats are not convincing, see the comments to the authors.
Comments to the authors:
- In fig.12 it is not clear how the threats affects the trajectory that is determined by the proposed algorithm. The regions with threats should be highlighted. The algorithms response on the same map with and without threats should be presented. If the black disks represents the threats then their positions does not affect the path planning. Thus the threats positions should be reconsidered.
- It is a good idea to refer at least one paper that is published in MDPI Biomimetics in order to show that the proposed manuscript matches the goals of the targeted journal.
Reviewer 2 Report
Comments and Suggestions for Authors
To address the problem of path planning for low-altitude penetration in complex environments, a photosensitivity-enhanced plant growth algorithm (PEPG) is proposed. Concerning recent results on UAV Path Planning, the authors might be aware of the so-called vector field method, such as recent result: Adaptation to unknown leader velocity in vector-field UAV formation. The relevance of this suggestion is that the vector field method can consider kinematic constraints of the UAV (as this manuscript), such as the turning angle. In addition, the suggestion shows path planning (as this manuscript), and a type of adaptation (adaptive control in the suggestion, photosensitivity-enhanced plant growth algorithm (PEPG) in this manuscript). The authors are free to judge if the suggestion above is appropriate. Concerning the methodology, the authors summarize it as
Based on the plant growth path planning algorithm (PGPP), the proposed algorithm improves upon the light intensity preprocessing and light intensity calculation methods. Moreover, the kinematic constraints of the UAV, such as the turning angle, are also considered. The planned path that meets the safety flight requirements of the UAV is smoother than that of the original algorithm, and the length is reduced by at least 8.2%.
The methodology is clear and the comparisons with the original algorithm are convincing. A few questions are listed
- The authors write 'With the rise and development of autonomy and intelligence technologies, UAV combat plays a pivotal role in the military field.' Personally, I would suggest to remove an explicit mention to military tasks, at least in the abstract. After all, UAVs have many civil applications and science should not have explicit military scopes
- the authors write 'simulation tests are carried out with three common path planning algorithms.' It is suggested that the abstract provide a clear mention to such algorithms
- the authors write 'The results show that the PEPG algorithm is superior to the other three algorithms in terms of the path length and path quality'. Path length is clear, but how to measure path quality?
- typo in Fig. 6 Obstcale
It is appreciated that an electronic chart of a sea area is selected as an example, cf. Figure 2. And the results are convincing. The results show that the PEPG algorithm is superior to the other three algorithms in terms of the path length and path quality. Overall, I am happy to support the work with the minor suggestions above
Reviewer 3 Report
Comments and Suggestions for Authors
The article proposes a method for solve the problem of low altitude penetration of UAVs to protect national territorial security. The problem is very interesting in the UAV path planning in the 2D and 3D environments. By the way, the photosensitivity-enhanced plant growth algorithm is encouraging to solve this problem. In addition, the presented results and analysis are inefficient for evaluation.
1. This paper gave me the impression that the introduction was too brief. I suggest expanding this section with related works which it is requested in paper, so that readers can understand the current research status of the problem, which this work focuses on. It is OK to write the contents about related works in the introduction, but I don't think introduction needs such a big space.
2. Re-write and include motivation, problem statement, remove short paragraphs.
3. Working algorithm is missing.
4. The section on tracking control test should be rewritten in a more readable manner. This section is very difficult to read.
5. Furthermore, it is strongly recommended to compare the proposed method with the PGPP algorithm. This paper proposes a method based on PGPP, so a comparison is necessary.
6. In conclusion, future work and limitations of the proposed algorithm should be discussed.
Round 2
Reviewer 3 Report
Comments and Suggestions for Authors
This manuscript can be accepted in this version.